# Association of DNA Promoter Methylation and *BRAF* Mutation in Thyroid Cancer

Farzana Jasmine [1,*], Briseis Aschebrook-Kilfoy [1,2], Mohammad M. Rahman [3], Garrett Zaagman [1], Raymon H. Grogan [4], Mohammed Kamal [5], Habibul Ahsan [1,2] and Muhammad G. Kibriya [1,2]

1   Institute for Population and Precision Health, Biological Sciences, University of Chicago, Chicago, IL 60637, USA
2   Department of Public Health Science, University of Chicago, Chicago, IL 60637, USA
3   Department of Pathology, Bangabandhu Sheikh Mujib Medical University, Dhaka 1000, Bangladesh
4   Department of Surgery, Baylor St. Luke's Medical Center, Houston, TX 77030, USA
5   Department of Pathology, The Laboratory, Dhaka 1205, Bangladesh
*   Correspondence: farzana@uchicago.edu

**Abstract:** The *BRAF* V600E mutation and DNA promoter methylation play important roles in the pathogenesis of thyroid cancer (TC). However, the association of these genetic and epigenetic alterations is not clear. In this study, using paired tumor and surrounding normal tissue from the same patients, on a genome-wide scale we tried to identify (a) any association between *BRAF* mutation and DNA promoter methylation, and (b) if the molecular findings may provide a basis for therapeutic intervention. We included 40 patients with TC (female = 28, male = 12) without distant metastasis. *BRAF* mutation was present in 18 cases. We identified groups of differentially methylated loci (DML) that are found in (a) both *BRAF* mutant and wild type, (b) only in *BRAF* mutant tumors, and (c) only in *BRAF* wild type. BRAF mutation-specific promoter loci were more frequently hypomethylated, whereas BRAF wild-type-specific loci were more frequently hypermethylated. Common DML were enriched in cancer-related pathways, including the mismatch repair pathway and Wnt-signaling pathway. Wild-type-specific DML were enriched in RAS signaling. Methylation status of checkpoint signaling genes, as well as the T-cell inflamed genes, indicated an opportunity for the potential use of *PDL1* inhibitors in BRAF mutant TC. Our study shows an association between *BRAF* mutation and methylation in TC that may have biological significance.

**Keywords:** thyroid carcinoma; *BRAF* mutation; methylation; RAS signaling; TNFR2

## 1. Introduction

Thyroid cancer is the most common malignant tumor of the endocrine system. Differentiated thyroid cancers are subdivided mainly into papillary thyroid cancer (PTC) and follicular thyroid cancer (FTC); PTC makes up 73.9%, and the follicular variant of papillary carcinoma (FVPTC) is the most common at 26.1% [1]. Most cases of the differentiated thyroid cancers have a good prognosis; the 5-year survival rate for thyroid cancer overall is 98.1% [2]. In a prospective study, after lobectomy in PTC, recurrence was found in 1.1% of patients after a median follow-up of 72 months [3]. Surgical removal (total or lobectomy depending on clinical evaluation) is the primary treatment for differentiated thyroid cancers. In differentiated thyroid cancer, the thyrocytes can pick up iodine and that is the rationale for post-surgical radio ablation with radioactive iodine (RAI) in select cases. For the RAI-refractory differentiated thyroid cancers, the multikinase or tyrosine kinase inhibitors may be used [4–8]. The activity of these multikinase inhibitors is not linked to specific genomic changes and may be related to the anti-angiogenic effects of these drugs. Some more specific FDA-approved drugs are also available targeting *BRAF* V600E, *RET*, and *TRK*, and have been reviewed in other papers [4,5,7,8].

The B-Raf proto-oncogene (*BRAF*) encodes a cytoplasmic serine/threonine kinase with a key role in regulating the mitogen-activated protein kinase signal transduction pathway [9]. The *BRAF* V600E mutation in exon 15 of the *BRAF* gene has been studied in different cancers, including thyroid cancers [9–20]. Induction of the *BRAF* V600E mutation in thyroid cells may lead to frequent hypermethylation [21]. In a meta-analysis, the overall prevalence of the *BRAF* mutation in thyroid cancer was found to be 45% [22]. *BRAF* mutation was found in 61.7% of PTC and in 1.7% of FTC [16]. The prevalence of *BRAF* V600E mutation was higher in conventional PTC (51.0%) than in FVPTC (24.1%) and FTC (1.4%) ($p$ = 0.0001) [23]. In patients with conventional PTC, *BRAF* V600E mutation was associated with older age, lymph node and distant metastasis, higher TNM stage, and recurrences [23]. In a study, *BRAF* V600E mutations were detected in 43.6% of papillary thyroid microadenoma and 42.4% of small PTC [18]. Of the mutant papillary thyroid microadenoma, 54.1% demonstrated aggressive characteristics as compared to 19.4% of the non-mutant microadenomas ($p$ = 0.010) [18]. In a review article, Ralph P. Tufano et al. showed PTC recurrence data of 2470 patients from 9 different countries [22]. The risk ratios in *BRAF*-mutation-positive patients were 1.93 ($p$ < 0.00001) for PTC recurrence, 1.32 ($p$ < 0.00001) for lymph node metastasis, 1.71 ($p$ < 0.00001) for extra thyroidal extension, and 1.70 ($p$ < 0.00001) for advanced stage [22]. Fine needle aspiration (FNA) biopsy is currently the best diagnostic tool for thyroid nodules [15], except for indeterminate or suspicious lesions found in 10–15% of cases, which remains a challenge [24].

DNA methylation, the most widely studied epigenetic mechanism, varies in different cancers, and is notably altered in thyroid cancer [25–32]. Now, the role of DNA methylation in cancer is being widely studied to determine markers that might guide treatment. Methylation markers may also detect different subtypes of thyroid cancer and augment early detection efforts [26]. The cancer genome is typically characterized by global hypomethylation concomitant with hypermethylation of the CpG island of the promoter regions associated with different cellular regulatory functions [33]. In recent years, thyroid cancer treatment is shifting towards personalized approaches to avoid over-diagnosis, overtreatment, and recurrences [34]. To investigate the methylation changes, a number of studies were conducted using the candidate gene approach utilizing amplicon sequencing [35,36], methylation-specific PCR [37–42], or qPCR [43,44]. There are also some genome-wide methylation studies using microarray on 27K BeadChip [45,46] or 450K BeadChip [29,47–50]. Methylation is also achieved on a genome-wide level using RRBS sequencing [51,52]. There are many genome-wide methylation studies where the fresh frozen thyroid tissue is used for microarray [29,49,50,53]. Zafon C et al. have summarized the DNA methylation studies in thyroid cancer [54]. Only a few have addressed the difference in methylation profiles between *BRAF* mutant and *BRAF* wild-type cases.

The *BRAF* V600E mutation is common in thyroid cancer [13,15,16,18,22,23,25], as is DNA methylation [25–32]. However, the interaction or association between them is not well-studied. In this study, we investigate the association of this genetic alteration (somatic mutation in *BRAF* V600E) and the epigenetic regulation (DNA promoter methylation) in the pathogenesis of thyroid cancer and explore the potential use of these molecular changes in identifying groups of patients for personalized treatment.

## 2. Materials and Methods

The study included a total of 40 consecutive patients ($M$ = 12, $F$ = 28) with histologically confirmed diagnosis of thyroid cancer from Bangladesh. Patients did not receive any radiotherapy or chemotherapy before surgery. Samples were collected from the operating room immediately after surgical resection. For each patient, one sample was obtained from the tumor mass, and another sample was taken from the resected, unaffected part of the thyroid and stored immediately at −86 °C. The histopathological diagnosis was carried out independently by two histopathologists at Bangabandhu Sheikh Mujib Medical University (BSMMU), Dhaka, Bangladesh. We also abstracted key demographic and clinical data and tumor characteristics for each patient from hospital medical records. The samples

were shipped on dry ice to the molecular genomics lab at the University of Chicago for subsequent DNA extraction and Methylation array.

### 2.1. DNA Extraction

DNA was extracted from fresh frozen tissue using the Gentra Puregene kit (Catalog# 158689, QIAGEN, Valencia, CA, USA). Quality control (QC) was performed for all samples using the Nanodrop 1000.

### 2.2. Genome-Wide Methylation Assay

The Infinium Human Methylation 450K BeadChip array was used to examine genome-wide DNA methylation (Illumina, San Diego, CA, USA). The methylation assay covered a total of 485,577 loci across the genome, of which 150,254 are on CpG island, 112,067 are on shore (0–2 kb from CpG island), 47,144 are on shelf (2–4 kb from CpG island) and the remaining 176,112 are in the deep sea (>4 kb from the CpG island). We excluded all the markers in sex chromosomes. The cross-tabulation of the methylation markers in the autosomes by functional group and in relation to CpG island is shown in Supplementary Table S1. In this study, we focused only on the promoter-associated markers in the CpG islands. For bisulfite conversion, EZ DNA methylation kit (Catalog# D5001, Zymo Research, CA, USA) was used. Paired samples (thyroid cancer and corresponding normal) were processed on the same chip at the same time to avoid the batch effect. The Illumina protocol was followed for the methylation assay. A Tecan Evo robot was used for automated sample processing and the chips were scanned on a single Illumina HiScan. Genome Studio version V2011.1 methylation module was used for data extraction.

### 2.3. BRAF and KRAS Mutation

Tumor and adjacent healthy thyroid tissue from 40 patients were tested for *BRAF* exon 15p.V600E mutation and *KRAS* (rs112445441) mutation by high-resolution melt analysis. We utilized the primers used previously by Gonzalez-Bosquet Jesus et al. [55]. Thermocycling and melting conditions were optimized for CFX96 instrument, and Bio-Rad Precision Melt Analysis software was used to identify *BRAF* positivity by differential melting curve characteristics.

## 3. Statistical Analysis

To compare the continuous variables (e.g., number of detected loci/samples or average signal intensity/average beta value etc. between the two groups), we used one-way analysis of variance (ANOVA).

**Genome-wide Methylation data analysis:** For measuring methylation, we used the Illumina Genome Studio software to generate the beta value for each locus from the intensity of methylated and unmethylated probes. The beta was calculated as (intensity of methylated probe)/(intensity of methylated probe + intensity of unmethylated probe). Hence, beta ranged between 0 (least methylated) and 1 (most methylated) and was proportional to the degree of methylated state of any particular loci. The beta values were exported to PARTEK Genomic Suite (https://www.partek.com/partek-genomics-suite/, accessed on 14 November 2022) for further statistical analyses. Principal component analysis (PCA) and sample histograms were checked as a part of the quality control analyses of the data. Mixed-model multi-way ANOVA (which allows for more than one ANOVA factor to be entered in each model) was used to compare the individual CpG loci methylation data across different groups. In general, "tissue" (tumor/adjacent normal) and sex (male/female) were used as categorical variables with fixed effect, since the levels "tumor/normal" and "male/female", represented all conditions of interest; whereas "case ID#" (as proxy of inter-person variation) was treated as a categorical variable with random effect, since the person ID was only a random sample of all the levels of that factor. Method of moments estimation was used to obtain estimates of variance components for mixed models [56]. As per the study design, we processed both the thyroid cancer tissue and the corresponding

adjacent normal sample from one individual in the same chip, and all the chips required to run the samples were run in a single batch to avoid batch effect. In the ANOVA model, the beta-value for the CpG loci was used as the response variable, and "tissue" (tumor or normal), case ID#, and "sex" were entered as ANOVA factors. One example of a model is as follows:

$$Y_{ijlm} = \mu + Tissue_i + Sex_j + CaseID_l + \varepsilon_{ijklm}$$

where $Y_{ijlm}$ represents the m-th observation on the i-th Tissue j-th Sex, l-th CaseID, $\mu$ is the common effect for the whole experiment, and $\varepsilon_{ijlm}$ represents the random error present in the m-th observation on the i-th Tissue j-th Sex l-th CaseID. The errors $\varepsilon_{ijlm}$ are assumed to be normally and independently distributed with mean 0 and standard deviation $\delta$ for all measurements.

In GO enrichment analysis, we tested whether the genes found to be differentially methylated fell into a Gene Ontology category more often than expected by chance. We used the chi-square test to compare "number of significant genes from a given category/total number of significant genes" vs. "number of genes on chip in that category/total number of genes on the microarray chip". A negative log of the *p*-value for this test was used as the enrichment score. Therefore, a GO group with a high enrichment score represented a lead functional group. The enrichment scores were analyzed in a hierarchical visualization and in tabular form.

In addition to looking at differential methylation at the level of individual CpG loci, we also examined differential methylation of a group of genes (gene set) using gene set ANOVA. Gene set ANOVA is a mixed-model ANOVA to test the methylation of a set of genes (sharing the same category) instead of an individual gene in different groups (https://www.partek.com/partek-genomics-suite/, accessed on 14 December 2022). The analysis is performed at the gene level, but the result is expressed at the level of the gene-set category by averaging the member genes' results. The equation for the model was:

$$\text{Model: } Y = \mu + T + P + G + S(T*P) + \varepsilon$$

where Y represents the methylation status of a Gene set-category, $\mu$ is the common effect or average methylation of the Gene set-category, T is the tissue-to-tissue (tumor/healthy) effect, P is the patient-to-patient effect, G is the gene-to-gene effect (differential methylation of genes within the GO-category independent of tissue types), S(T*P) is the sample-to-sample effect (this is a random effect, and nested in tissue and patient), and $\varepsilon$ represents the random error.

The characteristics of the thyroid cancer patients for this study (see Table 1) showed the majority of them had PTC (*n* = 29), followed by FVPTC (*n* = 7), and FTC (*n* = 4). A photomicrograph of each type is presented in Supplementary Figure S1. Patients with FTC were significantly older (age 55 years SD 7.07) than those with FVPTC (age 31.14 SD 8.47) or with PTC (age 35.89 years SD 15.41) (ANOVA *p* = 0.026). We did not have any patients with anaplastic carcinoma. Tumor size was larger in FTC (5.25 cm SD 2.87) compared to FVPTC (2.43 cm SD 1.90) or PTC (2.14 cm SD 0.95). *BRAF* V600E mutation was present in 18 (45%) cases. *KRAS* (rs112445441) mutation was not found in any of these tumors. Similarly to other studies, *BRAF* mutation was more common in PTC (55.2%) than FVPTC (28.6%), but the difference was not statistically significant (*p* = 0.40, Fisher's exact test). None of the FTC patients had *BRAF* mutation.

**Table 1.** Characteristics of the patients.

| Characteristics | Category | *BRAF* Mutation | *BRAF* Wild | *p*-Value |
|---|---|---|---|---|
| Sex | Female | 13 | 15 | 1 |
| | Male | 5 | 7 | |
| Age | Mean | 38.56 | 35.68 | 0.395 |
| | SD | 15.58 | 14.42 | |
| Pathology | PTC | 16 | 13 | 0.072 |
| | FVPTC | 2 | 5 | |
| | FTC | 0 | 4 | |
| Cervical lymph node | Absent | 12 | 15 | 1 |
| | Present | 6 | 7 | |
| Surgical procedure | Lobectomy | 2 | 6 | 0.258 |
| | Total thyroidectomy | 16 | 16 | |
| Tumor size (cm) | Mean | 2.11 | 2.82 | 0.54 |
| | SD | 1.16 | 1.92 | |
| Duration of illness (months) | Mean | 7.33 | 7.23 | 0.854 |
| | SD | 2.51 | 2.6 | |

### 3.1. DNA Promoter Methylation in Thyroid Cancer

We used paired (tumor and surrounding healthy thyroid tissues from the same individual) DNA samples to identify the differentially methylated loci (DML) in thyroid cancer. The magnitude of differential methylation (delta beta = beta value in tumor tissue—beta value in healthy thyroid tissue) may be different in the presence or absence of *BRAF* mutation. To identify DML for which the delta beta is statistically different in patients with or without *BRAF* mutation, we used an interaction term "tumor × *BRAF* mutation" in the ANOVA models. The *p*-values of this interaction term indicated that there were 4058 such DML in the promoter regions where the magnitudes of differential methylation were significantly different in *BRAF* mutant patients compared to *BRAF* wild-type patients. Examples from these loci with significant interaction *p* values are shown in Figure 1A,B. Figure 1A shows an example where the magnitude of differential methylation (hypomethylation) of *NHLRC4* gene in tumor tissue is greater in the presence of *BRAF* mutation compared to wild-type patients. Figure 1B shows an example when the magnitude of differential methylation of the *DDAH2* gene (hypermethylation) is greater in *BRAF* wild-type tumor tissue.

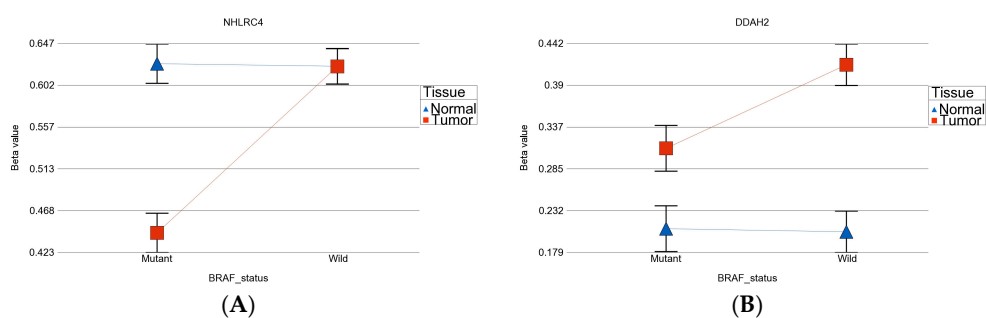

**Figure 1.** Examples of interaction between methylation status and mutation status. The left figure (**A**) shows larger magnitude of differential methylation (hypomethylation) in *BRAF* mutant tumor. The right figure (**B**) shows larger magnitude of differential methylation (hypermethylation) in *BRAF* wild type tumor.

In the next step, we analyzed the methylation data in *BRAF* mutant and in *BRAF* wild-type patients separately to identify DML in these two groups of patients. In the *BRAF* mutant group, there were 185 DML that were significant at FDR 0.05 level (see Figure 2A,B).

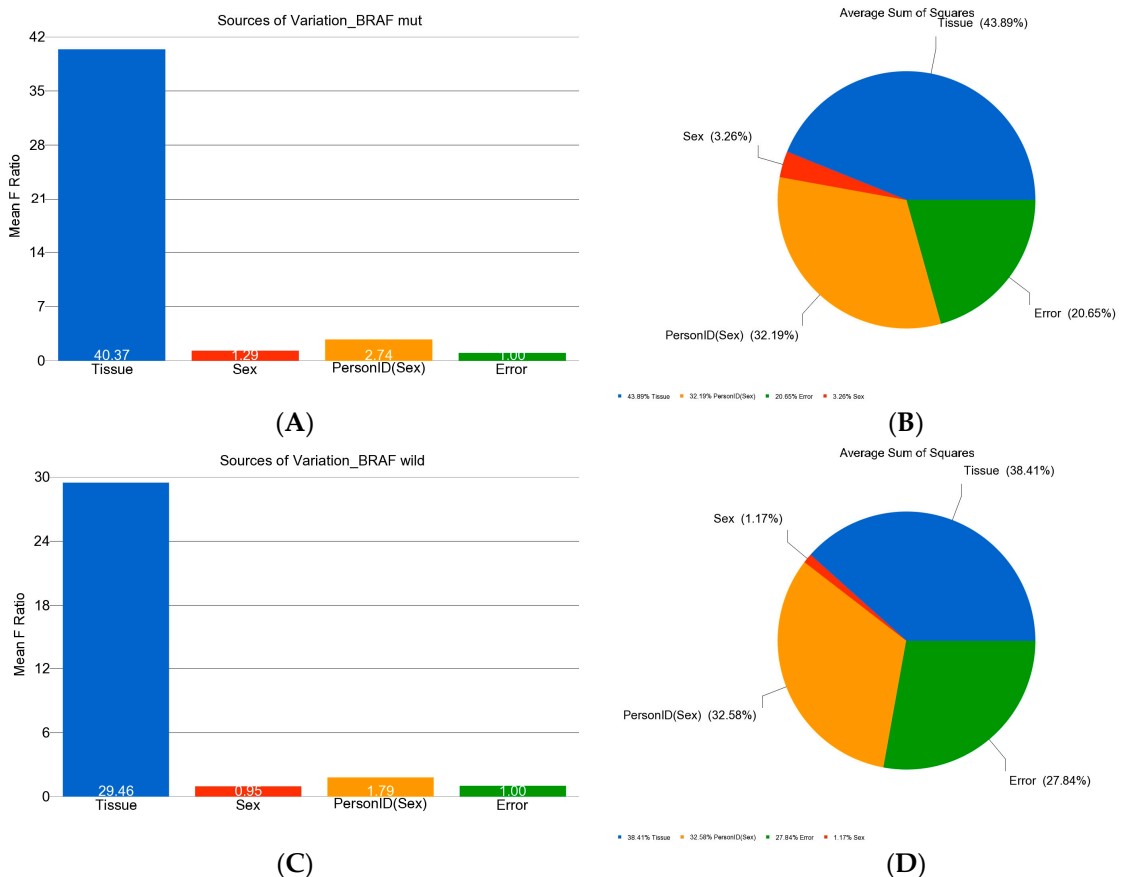

**Figure 2.** Variations of beta values in DML (at FDR 0.05) in patients with *BRAF* V600E mutation and wild type. The mean F-Ratio (F-statistics for the factor/the F-statistics for the model error) representing the significance of the factor in the ANOVA model are shown in the bar graph (**A**,**C**). The sum of squares in the ANOVA model representing the proportion of the variation explained by the factors is shown as the pie chart (**B**,**D**). The proportion of variation that can be explained by the "person-to-person variation" is shown in orange; the proportion that can be explained by the "tissue type (tumor or normal)" is shown in blue; the proportion that could not be explained by the ANOVA model (the "error") is shown in green. The variation in *BRAF* mutation is shown in the top panel (**A**,**B**) while the variation in the wild type is shown in the bottom panel (**C**,**D**).

In patients without *BRAF* mutation, there were 247 DML that were significant at FDR 0.05 level (see Figure 2C,D).

Figure 3 shows the overlap of these three lists of DML. Using the intersections of the Venn diagram, we identified the following three groups of DML:

1. DML (*n* = 51) common to *BRAF* mutation and wild type: These loci were differentially methylated in tumor tissue compared to healthy tissue, irrespective of *BRAF* mutation status, and the magnitude of the differential methylation was not different between *BRAF* mutant and wild-type tumors. Of these 51 DML, 34 were hypomethylated and 17 were hypermethylated in tumor tissue compared to corresponding normal thyroid tissue. An example of this group is shown in Figure 4A. The *CCND1* gene was hypomethylated in both groups. Reasonable separation of tumor and healthy tissue in the PCA plot using these 51 common loci for all the patients is shown in Figure 4B.

2. *BRAF* mutation-specific DML (*n* = 35): These loci were significantly differentially methylated in tumor tissue only in the presence of *BRAF* mutation, and the magnitude of differential methylation was significantly greater than in wild type. Of these 35 DML, 29 were hypomethylated and 6 were hypermethylated in tumor tissue compared to corresponding normal thyroid tissue (for detail, see Supplementary

Table S2). An example of this group is shown in Figure 4C. The *TNFRSF1B* gene (encoding *TNFR2*) was hypomethylated in the mutant group, suggesting potential overexpression of *TNFR2* in *BRAF* mutant tumors. This is an important gene in terms of therapeutic targeting potential, because *TNFR2* is rarely expressed in normal tissues. Reasonable separation of tumor and healthy tissue in the PCA plot, using these 35 *BRAF* mutation-specific loci for patients with *BRAF* mutation, is shown in Figure 4D.

3.  *BRAF* wild-type-specific DML (*n* = 62): These loci are significantly differentially methylated in tumor tissue only in the absence of *BRAF* mutation, and the magnitude of differential methylation is significantly greater than in patients with *BRAF* mutation. Of these 62 DML, 20 were hypomethylated and 42 were hypermethylated in tumor tissue compared to corresponding normal thyroid tissue. An example of this group is shown in Figure 4E. The *REC8* gene is hypermethylated in the *BRAF* wild-type group. Reasonable separation of tumor and healthy tissue in the PCA plot, using these 35 *BRAF* wild-type-specific loci for patients with wild-type *BRAF,* is shown in Figure 4F.

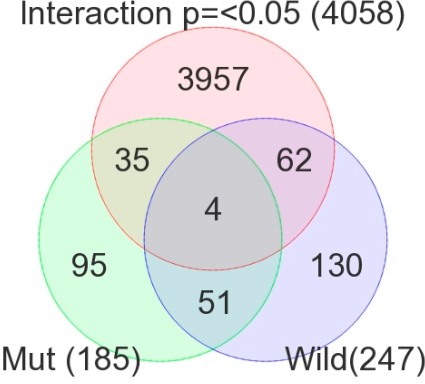

**Figure 3.** Overlap between the lists of DML (FDR 0.05) in *BRAF* mutant patients (light green), *BRAF* wild-type patients (light blue), and the methylation markers with significant ($p \leq 0.05$) interaction for "tissue x mutation" in ANOVA (in pink).

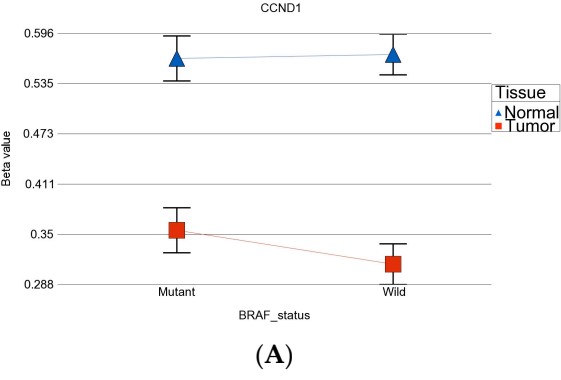
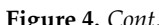

(**A**)

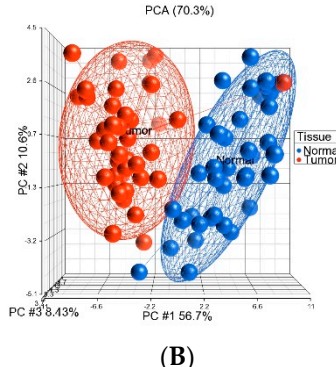

(**B**)

**Figure 4.** *Cont.*

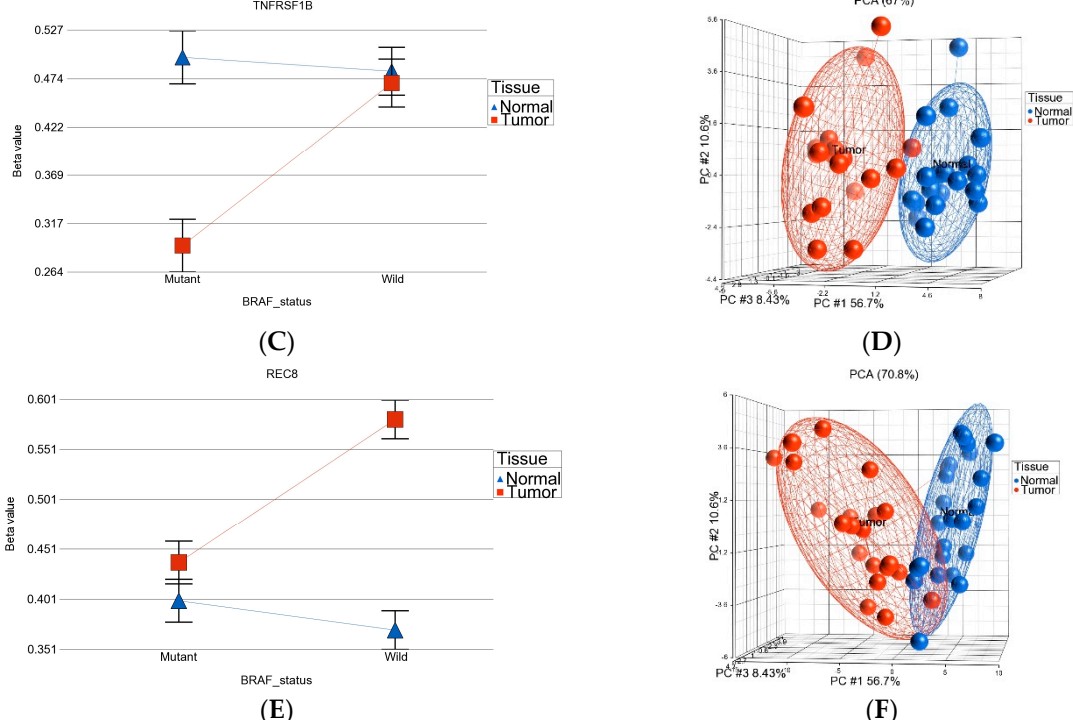

**Figure 4.** Three sets of DML in thyroid cancer. Examples of one locus from each set are shown on the left side (ACE), and the PCA plots from all the DML of each set are shown on the right side (**B**,**D**,**F**). The top panel (**A**,**B**) represents the 51 DML common in *BRAF* mutant and *BRAF* wild-type tumors, where (**A**) shows an example of this group—*CCND1* was significantly hypomethylated in tumor tissue compared to normal tissue, and the magnitude of differential methylation in *BRAF* mutant and wild-type was not different (interaction *p* > 0.05); (**B**) shows separation of the tumor tissue and normal tissue based on these common 51 DML. The middle panel (**C**,**D**) represents the *BRAF* mutation-specific 35 DML; the lower panel (**E**,**F**) shows the *BRAF* wild-type-specific 62 DML.

The methylation status of these three groups of DML shows that BRAF-mutation-specific promoter loci were more frequently hypomethylated (29 of 35), whereas BRAF wild-type-specific promoter loci were more frequently hypermethylated (42 of 62 DML) (*p* < 0.001, chi square test). GO-Enrichment analysis of the common 51 DML is shown in Figure 5, and the data are presented in detail in Supplementary Table S3. The result shows that this list of differentially methylated genes was enriched in genes related to a number of cancer-related KEGG pathways including mismatch repair pathway, Wnt-signaling pathway, pathways in cancer, pancreatic cancer, colorectal cancer, breast cancer, gastric cancer, and thyroid cancer. In fact, the *CCND1, RAC3,* and *FZD7* genes were common in many of these cancer pathways. We observed significant hypomethylation of the *CCND1* promoter region in thyroid tissue in both *BRAF* mutant and wild-type tumors, indicating potential over-expression of the *CCND1* gene in thyroid cancer (see Figure 6). However, we did not have gene expression data to confirm that.

We also looked at multiple promoter loci of the *CCND1* gene, which is shown in Figure 6.

GO-enrichment analysis of the list of 35 *BRAF* mutation-specific DML showed enrichment of genes related to auto-immune thyroid disease, type 1 diabetes, and human immunodeficiency virus (see Supplementary Table S4). *HLA-E* was the key gene. The differential methylation of all the probes in the promoter region of *HLA-E* gene (see Figure 7) shows that in *BRAF* mutant patients, *HLA-E* shows hypomethylation in tumor tissue, whereas in *BRAF* wild-type patients, the *HLA-E* shows slight hypermethylation in tumor tissue.

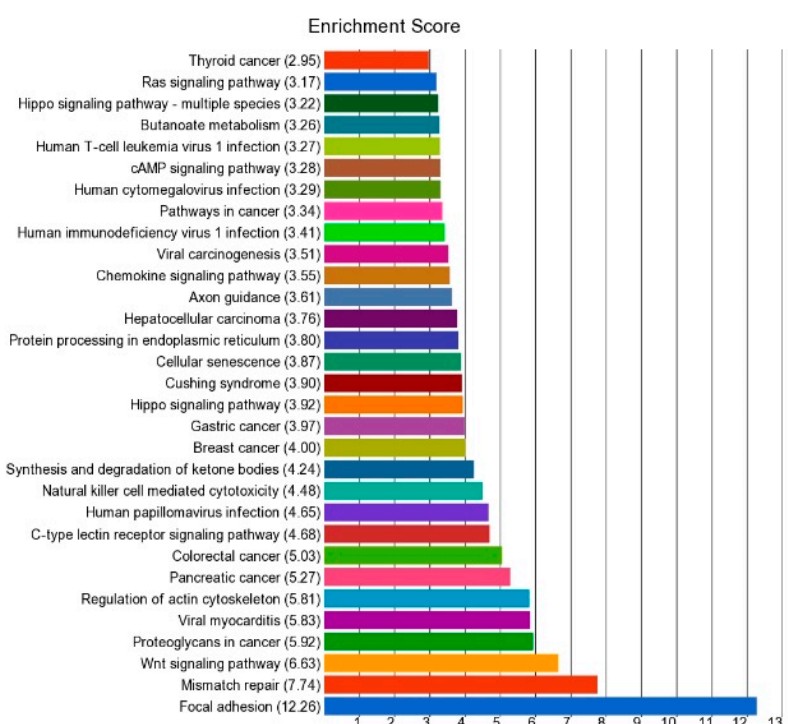

**Figure 5.** Enrichment score of the 51 common DML in thyroid cancer.

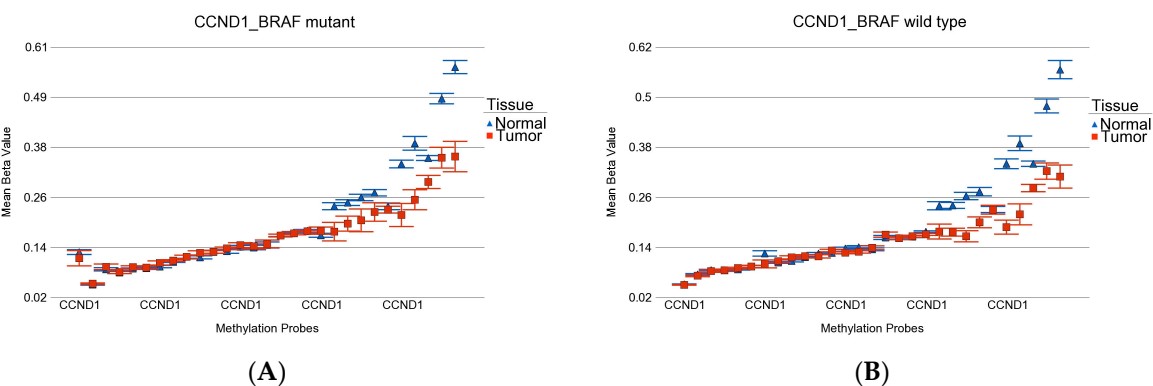

(**A**)                                           (**B**)

**Figure 6.** Differential methylation status of all the probes for *CCND1* promoter. Differential methylation in *BRAF* mutant patients is shown in (**A**), and the differential methylation in *BRAF* wild-type patients is shown in (**B**). Gene probes are arranged on the *x*-axis by methylation level, and the mean of the beta value is shown on the *y*-axis. Multiple probes show similar hypomethylation in both *BRAF* mutant and wild-type tumors.

GO-enrichment analysis of the list of 62 *BRAF* wild-type-specific DML showed enrichment of genes related to the Ras signaling pathway (see Supplementary Table S5). *RASSF1* is one of the key players (see Figure 8). Differential methylation of *RASSF1* in cancer is demonstrated in other studies [57]. Another interesting enrichment was observed in genes related to Yersinia infection (*ARF6*, *IRF3*, *GIT2*), which is usually seen with consumption of unpasteurized milk or raw pork.

We also looked at the methylation status of some genes that have been reported to be differentially methylated in thyroid cancer in previous studies, **as** shown in Supplementary Table S6 [38,57–61]. The differential methylation of these genes in patients with *BRAF* mutation and wild type are presented in Supplementary Tables S7 and S8, respectively.

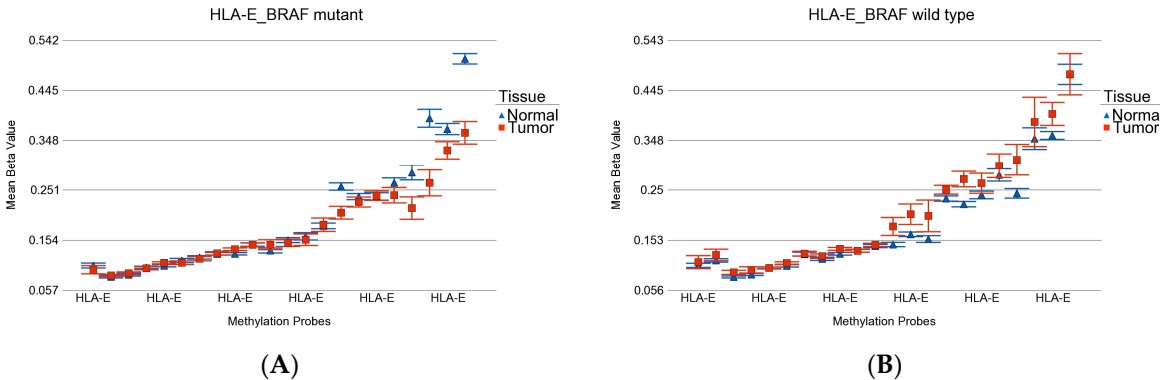

**Figure 7.** Differential methylation status of *HLA-E*. Differential methylation in *BRAF* mutant patients is shown in (**A**), and the differential methylation of *BRAF* wild-type patients is shown in (**B**). Note that multiple probes were hypomethylated in *BRAF* mutant tumor and hypermethylated in *BRAF* wild-type tumor.

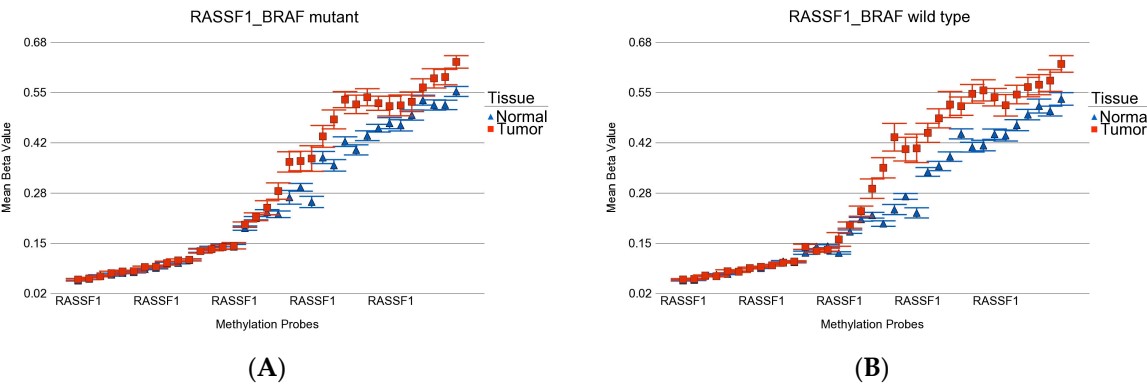

**Figure 8.** Differential methylation status of *RASSF1*. Differential methylation in *BRAF* mutant patients is shown in (**A**), and the differential methylation of *BRAF* wild-type patients is shown in (**B**). Multiple probes were hypermethylated in both groups. However, the overall magnitude was slightly higher in wild type.

In the next step, instead of individual gene probes, we examined whether a set of genes (sharing the same category or functional group) was differentially expressed in thyroid cancer tissue compared to healthy skin tissue. We used the gene set ANOVA.

Among the cancer-related gene sets (see Supplementary Table S9 for the list), the tumor suppressor genes were overall hypermethylated in both *BRAF* mutant (ANOVA $p = 0.006$) and *BRAF* wild-type (ANOVA $p = 2.53 \times 10^{-7}$) patients (see Figure 9), but the magnitude of differential methylation was greater in wild type (ANOVA interaction $p = 0.038$). This hypermethylation may cause more downregulation of tumor suppressor genes in *BRAF* wild type.

Among the DNA damage-related gene set (see Supplementary Table S10 for the list), the checkpoint signaling genes were hypermethylated in both *BRAF* mutant (ANOVA $p = 0.0003$) and *BRAF* wild-type (ANOVA $p = 4.91 \times 10^{-28}$) patients (see Figure 10), but the magnitude of differential methylation was greater in wild type (ANOVA interaction $p = 6.41 \times 10^{-10}$). This may be important from a therapeutic perspective regarding the potential efficacy of immune checkpoint inhibitors (ICI).

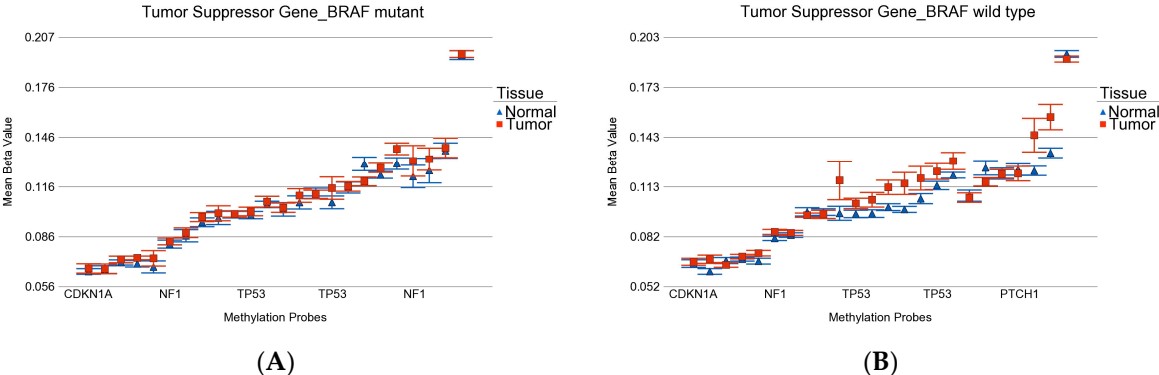

**(A)** **(B)**

**Figure 9.** Differential methylation status of tumor suppressor genes. Differential methylation in *BRAF* mutant patients is shown in (**A**), and the differential methylation on *BRAF* wild-type patients is shown in (**B**). Gene probes are arranged on the *x*-axis by methylation level, and the mean of beta value is shown on the *y*-axis. For many genes, there were multiple probes on the chip. Gene symbols for all of the gene probes could not be shown on the *x*-axis. Multiple probes were hypermethylated in both groups. Nevertheless, the overall magnitude was slightly higher in wild type.

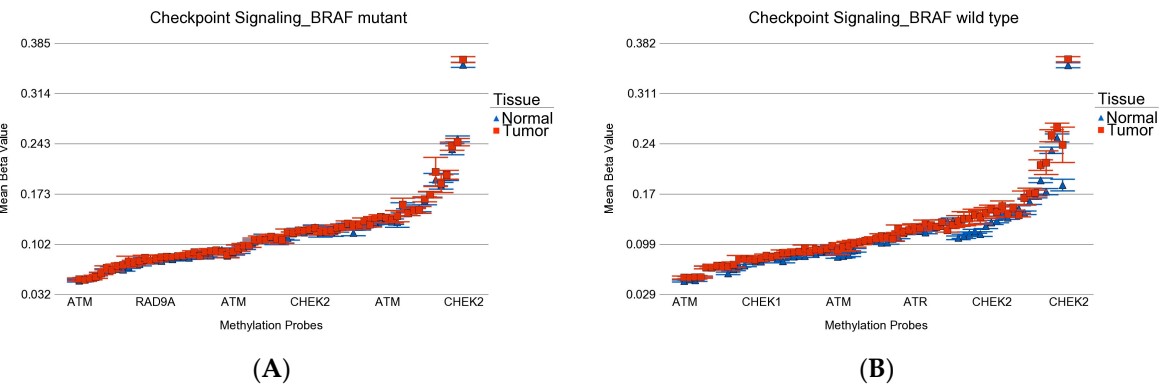

**(A)** **(B)**

**Figure 10.** Differential methylation status of checkpoint signaling genes. Differential methylation in *BRAF* mutant patients is shown in (**A**), and the differential methylation of *BRAF* wild-type patients is shown in (**B**). Gene probes are arranged on the *x*-axis by methylation level, and the mean of beta value is shown on the *y*-axis. For many genes, there were multiple probes on the chip. Gene symbols for all gene probes could not be shown on the *x*-axis. Multiple probes were hypermethylated in both groups. However, the overall magnitude was slightly higher in wild type.

*T-cell inflamed gene*: This group of genes (see Supplementary Table S11 for the list) has been used to predict immunotherapy targeting programmed cell death protein-1 (PD-1, also known as CD274)) [62,63]. We looked at the differential methylation of these genes in our patients to see if they were different by *BRAF* mutation status (see Figure 11A,B). We observed that these genes were slightly but significantly hypomethylated in patients with *BRAF* mutation ($p = 3.94 \times 10^{-9}$), whereas in *BRAF* wild-type patients, these genes were hypermethylated ($p = 4.45 \times 10^{-7}$). This may suggest that from a gene expression point of view (for which we do not have data), these genes may be overexpressed in tumor tissue in the presence of *BRAF* mutation and may be a better candidate for *PDL1* inhibitors.

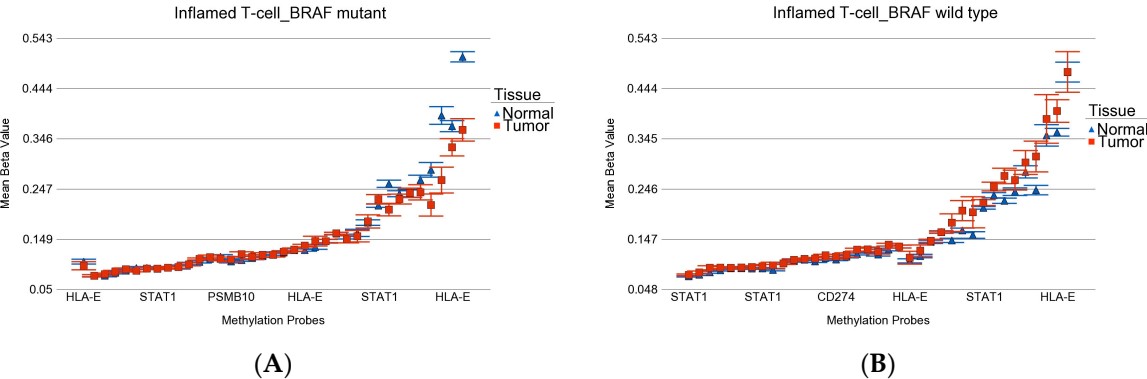

(**A**)  (**B**)

**Figure 11.** Differential methylation status of inflamed T-cell genes. Differential methylation in *BRAF* mutant patients is shown in (**A**), and the differential methylation of *BRAF* wild-type patients is shown in (**B**). Gene probes are arranged on the *x*-axis by methylation level, and the mean of beta value is shown on the *y*-axis. For many genes, there were multiple probes on the chip. Gene symbols for all gene probes could not be shown on the *x*-axis. Note that multiple probes were hypomethylated in *BRAF* mutant tumor and hypermethylated in *BRAF* wild-type tumor (interaction $p = 3.67 \times 10^{-13}$).

### 3.2. Validation of the DML in Independent Set of Samples

Given the fact that this current study was conducted in a Bangladeshi population, for validation purposes, we used the methylation data from our previous study [29], where all the samples (16 PTC thyroid tumors and 13 healthy thyroid tissues) were collected from US patients and the same chip was used for methylation analysis. We pulled the beta value of all the 51 common DML that we described in this study, of which 49 markers were also significantly differentially methylated in the US patients. (see Figure 12)

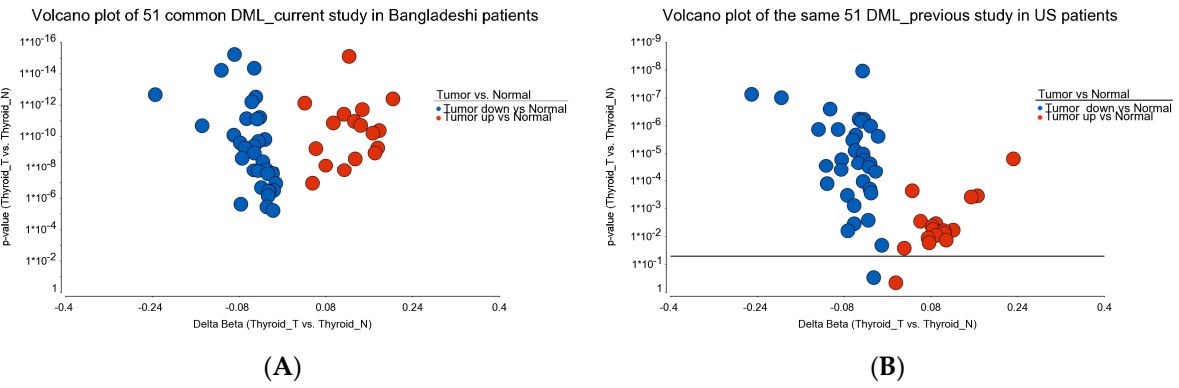

(**A**)  (**B**)

**Figure 12.** Venn diagram of the 51 common DML found in the current study. The differential methylation of these loci in Bangladeshi patients (current study) is shown on the left (**A**); the differential methylation of these same 51 loci in US patients (previous study) is shown on the right (**B**).

### 4. Discussion

In our study, the *BRAF* V600E mutation was present in 18 (45%) cases, and it was more common in PTC (55.2%) than FVPTC (28.6%). Our result is consistent with previous studies on *BRAF* mutation and thyroid cancer [16,23]. We investigated the differential methylation of genes using 450 K methylation array of paired thyroid tumor and normal tissue from the same individuals with and without *BRAF* mutation. We identified groups of genes that were (a) differentially methylated irrespective of *BRAF* mutation status, (b) differentially methylated only in *BRAF* mutant tumors, and (c) differentially methylated only in *BRAF* wild-type tumors. BRAF mutation-specific promoter loci were more frequently hypomethylated, whereas BRAF wild-type-specific promoter loci were more frequently hypermethylated. Some genes, such as the *NHLRC4* gene (ubiquitin

protein ligase activity) showed differential methylation (hypomethylation) in tumor tissue that was greater in the presence of *BRAF* mutation compared to wild-type patients. The *DDAH2* gene showed more differential methylation (hypermethylation) in tumor tissue in *BRAF* wild-type tumors compared to the mutant group. Huang S et al. found that *DDAH2* protein may be associated with blocking the activation of the innate immune response [64]. In *BRAF* wild-type tumors, the hypermethylation of *DDAH2* may be associated with lower expression of *DDAH2* and may have a beneficial effect.

We found that *CCND1* was significantly hypomethylated, both in the presence and absence of *BRAF* mutation. This finding is consistent with gene expression data from other studies. Sora Jeon et al. showed that cyclin D1 is consistently overexpressed in PTC, and cyclin D1 immunostaining is useful for identifying the extent of tumor involvement [65]. The cyclin D1 protein, coded by the *CCND1* gene, is a gate-keeper regulating the transition from the G1 phase into the S phase of the cell cycle [66]. Overexpression of cyclin D1 is observed in a variety of human cancers, and is involved in tumorigenesis [66]. The overexpression of cyclin D1 in human cancers can result from genetic alterations, changes in epigenetic regulation, gene transcription, and protein translation of *CCND1*. Expression of cyclin D1 protein was observed at varying levels in 18/27 anaplastic thyroid cancers (67%) [67]. Our study suggests that in thyroid cancer, hypomethylation of the *CCND1* promoter may be the cause of *CCND1* overexpression seen in other studies.

TNF is a unique cytokine that exerts two distinct actions depending on its two receptors; one is tumor necrosis factor receptor 1 (TNFR1), which is regulated by *TNFRSF1A* gene, and the other is TNFR2, regulated by *TNFRSF1B.* The TNFR1 surface receptor is linked to a cell death pathway, while TNFR2 is linked to a cell proliferation pathway. *TNFR2* is a particularly important molecular target because it is rarely expressed in normal tissues and is overexpressed in many types of cancer cells and tumor microenvironments [68]. In a recent review, Takahashi et al. have discussed the role of TNFR2 signaling in cancer [68]. TNFR2 is overexpressed in a variety of cancers, and its activation promotes tumor growth and progression. TNFR2 is a signaling molecule found on the surface of a subset of potent regulatory T cells (Tregs) that can activate the proliferation of these cells through nuclear factor kappa B (NF-kB) [69]. TNFR2 is an attractive target protein because of its restricted abundance in highly immunosuppressive Tregs and oncogenic presence in human tumors [70,71]. Therefore, the targeting of malignant cells with TNFR2-specific antagonistic antibodies may not only control cancer growth but also minimize adverse effects [71]. In our study, we found that in tumor tissue from the *BRAF* mutant group, *TNFRSF1B* (encoding TNFR2) was significantly hypomethylated, suggesting potential overexpression of TNFR2 in this group of patients. We admit our limitation in that we do not have gene expression data in this study. However, to our knowledge, no study has shown the differences in *TNFRSF1B* (encoding TNFR2) methylation (and thereby expression of TNFR2) dependent on *BRAF* mutation status in thyroid cancer. Furthermore, no studies have thus far directly investigated DNA methylation/demethylation of TNFR2 in malignant disease. Recently there has been advancement in TNFR2-targeted immunotherapy [68,69,71]. If our finding is confirmed by gene expression data in a future study, then one may consider investigating the effect of TNFR2-targeted antibody therapy in BRAF mutant thyroid cancer patients in the case of RAI failure.

As per current guidelines from the National Comprehensive Cancer Network (NCCN) [4] and consensus statement from the American Head and Neck Society Endocrine Surgery Section and International Thyroid Oncology group [7], only patients with progressive or symptomatic RAI-resistant thyroid cancer should be considered for medical therapy; that may be guided by genetic testing [8], such as BRAF V600E mutation (for BRAF &/or MEK inhibitor), RET fusion (for RET inhibitors), NTRK1/2 fusion (for NTRK inhibitors), or ALK fusion (for ALK inhibitor) [4,7]. Epigenetic testing has a role in diagnostics. However, epigenetic testing is currently of limited utility in therapeutic decision making, as there is currently no commercially available compound for gene-specific methylation modification. Therefore, its potential utility in therapeutics may be indirect, as a proxy for predicting gene

expression. In this context, methylation data from our current study indirectly suggests an opportunity for potential use of PDL1 inhibitor in BRAF mutant thyroid cancer.

In our study, *REC8* showed significant hypermethylation in tumor tissue for the *BRAF* wild-type group when compared to the *BRAF* mutant group. *REC8* has tumor suppressor activity. The role of the PI3K pathway in human cancer has been well established. The role of *REC8* in this PI3K pathway was studied by Liu et al. [72]. They showed that *REC8* (classically known as a meiotic-specific gene) is robustly downregulated by the PI3K pathway through hypermethylation. *REC8* hypermethylation was strongly associated with genetic alterations and activities of the PI3K pathway in thyroid cancer cell lines, thyroid cancer tumors, and some other human cancers. Their findings suggest that *REC8* may have tumor suppressor effects and act as a robust epigenetic target of the PI3K pathway. Aberrant inactivation of *REC8* through hypermethylation by the PI3K pathway may represent an important mechanism for mediating the oncogenic functions of the PI3K pathway.

The *RASSF1* gene showed hypermethylation in tumor tissue in both *BRAF* mutant and wild-type cases. However, the overall magnitude was slightly higher in wild type. In a study of PTC, correlations between the methylation status of four genes (*TIMP3, RASSF1A, RARβ2,* and *DCC*) and the *BRAF* V600E mutation were studied. *RASSF1A* methylation decreased the probability of *BRAF* mutation, while methylation in other genes increased the probability of *BRAF* mutation [57].

When we analyzed the methylation status of *HLA-E*, we found that multiple probes were hypomethylated in *BRAF* mutant tumors and hypermethylated in *BRAF* wild-type tumors. We could not find the gene expression data for *HLA-E* and thyroid cancer with *BRAF* mutation in other studies, but *HLA-E* and *HLA-F* expression significantly correlated with depth of invasion, nodal involvement, lymphatic invasion, and venous invasion in gastric cancer patients [73].

There are a few studies showing differences in methylation and/or gene expression in relation to *BRAF* mutation. In PTC, 38% of cases exhibit a complete lack of *SMOC2* expression, which is attributed to the presence of *BRAF* V600E mutations [74]. The results of the analysis of DNA methylation chip indicate that the *SMOC2* gene initiator region contains a high-methylation CpG site [74]. A significant association was observed between thyroid stimulating hormone receptor (*TSHR*) gene methylation and positive *BRAF* V600E mutation cases in thyroid cancer [74]. In thyroid cancer with *BRAF* gene mutations, the presence of promoter methylation in *SMOC2, TSHR, TERT, SLC5A8, PLEKHS1, PTEN, DAPK, PDLIM4,* and *RSK4* genes would lead to poor prognosis of thyroid cancer [74]. *PTGS2, HOXA1, TMEFF2, p16,* and *PTEN* genes were hypermethylated in FNAC of thyroid tumor when compared between the tumor and healthy tissue. There were no significant differences in the methylation status of these genes between *BRAF* mutation negative and positive cases [58].

A cell line study was performed using a 12K methylation microarray on two cell lines harboring *BRAF* mutation after knockdown of *BRAF, HLX1, KLHL14, HMGB2, NR4A2, FGD1,* and *ZBTB10* genes. *BRAF* gene knockdown was associated with hypermethylation of about 59 genes, and all were related to cell cycle, cell development, DNA replication, and inflammatory response [75].

One of the major limitations of our study is the lack of gene expression data from the same tissue, which would allow us to confidently describe some of the methylation results in terms of downstream effects. Depending on funding, we plan to do that in the future. We also do not have post-surgical clinical follow-up data to associate our findings with prognosis. We are also limited with only 18 BRAF mutant and 22 wild-type cases. With all of the limitations of our study in mind, some of the strengths of the study may be noted. First, paired tumor-normal samples from the same individual for comparison is the most robust method for detecting any methylation changes in cancer. Second, we used properly preserved fresh frozen tissue samples for methylation analysis. An array-based methylation study from FFPE samples is difficult [76]. Third, to our knowledge, this is one of the first studies on native Bangladeshi patients with thyroid cancer to comprehensively

study a difference of methylation between *BRAF* wild-type and *BRAF* mutant thyroid cancer patients. Fourth, our study also shows that this methylation data may indicate a molecular basis for certain therapeutic options for a specific group of patients.

## 5. Conclusions

DNA methylation plays an important role in the pathogenesis of thyroid cancer. Many genes are differentially methylated irrespective of *BRAF* V600E mutation status; however, there are also *BRAF*-mutation-specific genes that show a markedly different magnitude of methylation in thyroid cancer tissue. Some of the common and mutation-specific DNA methylation data may have clinically relevant biological significance and potential therapeutic implications for precision medicine.

**Supplementary Materials:** The following supporting information can be downloaded at: https://www.mdpi.com/article/10.3390/curroncol30030227/s1. Figure S1. Photomicrograph of Thyroid carcinoma. Table S1: Cross-tabulation of Regulatory features and Relation to CpG Island; Table S2: *BRAF* mutation-specific 35 DML; Table S3: Enrichment Score_Common 51 DML; Table S4: Enrichment Score of *BRAF* mutation-specific 35 DML; Table S5: Enrichment Score_of *BRAF* wild-type-specific 62 DML; Table S6: DML in thyroid cancer from earlier studies; Table S7: Differential methylation of previously reported genes in *BRAF* mutant; Table S8: Differential methylation of previously reported genes_in *BRAF* wild; Table S9: List_of Cancer-Related Genes; Table S10: List of DNA damage genes; Table S11: List of T-cell inflamed Genes.

**Author Contributions:** Conceptualization, F.J., H.A. and M.G.K.; methodology, M.M.R., M.K. and F.J.; formal analysis, M.G.K.; investigation, M.M.R., M.K. and F.J.; resources, H.A., B.A.-K. and R.H.G.; data curation, F.J. and M.G.K.; writing—original draft preparation, F.J., G.Z. and M.G.K.; writing—review and editing, B.A.-K., R.H.G. and H.A.; supervision, M.K., H.A. and M.G.K.; funding acquisition, H.A. All authors have read and agreed to the published version of the manuscript.

**Funding:** This study was partially supported by NIH funds P20CA210305.

**Institutional Review Board Statement:** The research protocol was approved by the ethical review committee of BSMMU, Dhaka, Bangladesh (BSMMU/2012/6407) and by the IRB of the University of Chicago (13-1104), IL, USA.

**Informed Consent Statement:** Informed consent was obtained from all subjects involved in the study.

**Data Availability Statement:** All supporting data are presented in the tables presented in the main manuscript and as Supplementary Material.

**Acknowledgments:** We acknowledge the support and help from all of the patients included in this study. We thank the University of Chicago Research, Bangladesh (URB) staff for the handling and shipping of all of the study material to the University of Chicago molecular genomics laboratory. We also thank Anthony Quinn and Aaron Munoz for editing the manuscript.

**Conflicts of Interest:** The authors declare no conflict of interest. The funders had no role in the design of the study; in the collection, analyses, or interpretation of data; in the writing of the manuscript; or in the decision to publish the results.

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
