# Peer review of "Association of DNA Promoter Methylation and BRAF Mutation in Thyroid Cancer"

_curroncol, doi:10.3390/curroncol30030227_

Round 1

Reviewer 1 Report (Previous Reviewer 3)

Your revisions more than adequately answer my concerns with your original version. The paper now reads well and should be of interest to al those involved in the management of differentiated thyroid cancer.

I would though advise you include the word "differentiated" in line 48, as this statement would not be true for all thyroid cancer (given the poor outlook from anaplastic carcinoma particularly).

Reviewer 2 Report (Previous Reviewer 2)

Thank you for addressing the comments. I have reviewed the responses and I am convinced with the changes made in the updated version of the article. I have no additional comments.

This manuscript is a resubmission of an earlier submission. The following is a list of the peer review reports and author responses from that submission.

Round 1

Reviewer 1 Report

Authors set out to investigate the association of this genetic alteration (somatic mutation in BRAF V600E) and the epigenetic regulation (DNA promoter methylation) in the pathogenesis of thyroid cancer.

There were 29 patients with PTC and 7 with FVPTC; BRAF V600E mutation was present in 18 (45%) cases. There were 18 BRAFmt across 40 patients in the total cohort; 16 BRAFmt in 29 PTCs and 2 in FVPTC.

There is no investigation or attempt to understand if methylation changes translate to mRNA expression or protein abundance. This major deficit prohibits statements as in line 21/22 of the abstract “Among the BRAF mutation-specific genes, TNFRSF1B (encoding TNFR2) hypomethylation is clinically important as it is rarely expressed in normal tissues and thus makes this very important molecular target in this BRAF mutant TC”. The study does not provide any data about TNFR mRNA or protein expression, does not identify this as a molecular target, and we cannot know if –or not – it is clinically important. The same criticism applies to the other findings.

The data are of some interest – but the analysis lacks the depth to make sense of the observations which may in part be spurious and certainly without meaningful explanation. Statistical association forbids speculation about causation. Indeed, when testing several thousand methylation sites one can be assured to find statistical outliers, yet their significance remains unknown.

Figure 5 shows enrichment in Wnt and focal adhesion pathways. These can be meaningfully linked to BRAF mutations (e.g. Chen et al Wnt/β-Catenin Pathway Activation Mediates Adaptive Resistance to BRAF Inhibition in Colorectal Cancer. Mol Cancer Ther. 2018 Apr;17(4):806-813. doi: 10.1158/1535-7163.MCT-17-0561. Epub 2017 Nov 22. PMID: 29167314) – and such links should have been discussed/explored more fully.

The article would much benefit from re-writing in a more concise style.

Reviewer 2 Report

In the current manuscript entitled "Association of DNA Promoter Methylation and BRAF mutation in Thyroid Cancer" by Farzana Jasmine et al, the authors have described on BRAF V600E mutation and DNA promoter methylation playing pivotal role in Thyroid Cancer. In this study, the authors have used paired tumor and surrounding normal tissue from same patients to identify groups of differentially methylated loci in BRAF mutant and WT tumors and found several genes that show significant interaction of BRAF mutation and methylation in Thyroid Cancer. This study has great potential in the field as it provides a lot of data from clinical patients. It would be appropriate for the authors to improve the manuscript as per the following comments to improve its quality for publication:

Lines 112-113: Provide catalog number of the kit.

Line 124: Provide catalog number of the kit.

Figure 1: Move the figure legend on the right side of the graphs

Figure 2: Increase the font size of the figure legends and bring them on the right side of the figure so these are clearly visible, and readers can find them appropriately

Figure 3: 'Interact' or 'Intact'. Please check the term.

Line 257: Correct the term 'mutant'

Figure 4: Move the figure legend on the right side of the graphs

Figure 6: Move the figure legend on the right side of the graphs

Figure 7: Move the figure legend on the right side of the graphs

Figure 8: Move the figure legend on the right side of the graphs

Figure 9: Move the figure legend on the right side of the graphs

Figure 10: Move the figure legend on the right side of the graphs

Figure 11: Move the figure legend on the right side of the graphs

Figure 12: Increase the font size of the figure legends and bring them on the right side of the figure so these are clearly visible, and readers can find them appropriately

Please check for any typos and errors in the manuscript. In addition, it would be more appropriate if these figures could be made using GraphPad Prism software, and that these look more appropriate (figure legends on right side, and appropriate font size of legends, axis, title etc.).

Reviewer 3 Report

The basic core of this paper appears very sound but needs a more solid Introduction to help define the place of your findings in clinical practice.

A number of the references in the Introduction are not that recent. As it is well recognised that differentiated thyroid cancer (DTC) is becoming more common, you need to cite more recent publications when discussing subtypes and outcome (refs 2, 3 and 8 need replacing). I am not sure how discussing treatment options is relevant to the paper as a whole but chemotherapy is little used now (I would omit this and reference 5) as targeted therapy is now standard of care for patients who become iodine-refractory (rather than "have been proposed" line 44). The phrase "At present there is no successful treatment" is incorrect. In this context not curable and not treatable mean something very different. 

Similarly in the paragraph on BRAF mutation. There are lot of more up to date references (i.e. in the last 5 years) on this topic. There is contradiction in this paragraph where in lines 53-54 and in lines 58-64, you describe BRAF mutation being associated with a higher risk of recurrence but then in lines 66-68, not being a risk factor. 

In lines 69-74, you introduce DNA methylation but do not explain how this impacts clinical outcome. In what way does this affect prognosis? Does hypermethylation affect this differently from hypomethylation? To what extent is this gene-dependent? In line 91, you state that "BRAF mutation is common in thyroid cancer as is DNA methylation" but give no reference to support this. Only in the Discussion do you mention your previous work in this area (Ref 32). It would be appropriate to include this in the Introduction. 

In lines 98-110, you describe the origin of the material studied. Only the principles need to go in this section. The details of the patients belong in the Results section (where they are reported in lines 194-196).

Figure 3: the legend should explain what the Interact group represents.

The Discussion section is good and reads well, the paragraph on limitations of this study being particularly helpful.